# Concerns and Strengths: Caregiver Perceptions of Their Infant/Toddler with Prenatal Alcohol Exposure [note 1]

**DOI:** 10.3390/children10030544

**Published:** 2023-03-13

**Authors:** Misty Pruner, Tracy Jirikowic, Carolyn Baylor, Susan Astley Hemingway

**Affiliations:** 1Center on Human Development and Disability, University of Washington, Seattle, WA 98195, USA; 2Department of Rehabilitation Medicine, University of Washington, Seattle, WA 98195, USA; 3Department of Epidemiology, Department of Pediatrics, University of Washington, Seattle, WA 98195, USA

**Keywords:** prenatal alcohol exposure, fetal alcohol spectrum disorder, infant, toddler, strengths, concerns

## Abstract

Caregiver-reported assessments provide opportunities for caregivers to share concerns and identify the strengths of their infant/toddler regarding prenatal alcohol exposure (PAE). These insights may reveal under-recognized concerns and inform a strengths-based approach to early intervention. The purpose of this study was to describe the type and frequency of caregiver-reported concerns and strengths in a sample of infants/toddlers at the time of their fetal alcohol spectrum disorder (FASD) diagnostic evaluation. Caregivers’ concerns and strengths were identified in the context of two parent-report questionnaires, the Infant Toddler Sensory Profile and Child Behavior Checklist/1½-5. By using content analysis, caregivers’ open-ended responses were identified, coded, and analyzed. The frequencies of all the coded concerns and strengths were counted. The data were compared across the two age groups (<2 years and ≥2 years) and caregiver status. Caregivers (*n* = 117) identified numerous concerns and strengths across multiple categories. The most frequently reported concerns were related to aggressive behavior, language/communication, and sensory processing. The most frequently reported strengths were related to happiness, sociability, and love. The type of concerns and strengths reported were relatively consistent across age and caregiver status. These findings reinforce the value of caregivers’ perspectives and offer a reminder to practitioners that infants/toddlers with PAE and their caregivers have many strengths that can be harnessed, in addition to a range of challenges that must be addressed.

## 1. Introduction

Prenatal alcohol exposure (PAE) can disrupt the neurodevelopmental and behavioral trajectory of infants/toddlers with lasting impacts on learning, mental health, and overall well-being [1,2]. Fetal alcohol spectrum disorder (FASD) is a term used to describe the full range of physical, cognitive, and behavioral impairments caused by PAE, which are estimated to occur in at least 1% of children and youth in the general population [3,4]. Infants/toddlers with PAE are a heterogeneous group of children who may experience a wide range of delays in development, sensory processing, and/or emotional and behavioral functioning [5]. Challenges in any one of these developmental domains can limit participation in everyday routines and activities and negatively influence the quality of parent–child interactions and early relationships [6,7]. Conversely, infants/toddlers with PAE also possess individual strengths and positive attributes [8] that can serve as protective factors and support whole child development. 

Early interventions that target risk factors and build on individual strengths can alter the course of development in a positive direction. Findings from decades of developmental and intervention science have demonstrated the substantial benefits of early intervention on child development and family well-being [9]. The first three years of life have been recognized as an incredibly important time for child development, given the brain’s capacity for change and sensitivity to environmental influences [10]. With an emphasis on promoting healthy parent–child interactions and strengthening family adaptation, a family-centered early intervention approach is well-suited to respond to the diverse needs of infants/toddlers with PAE [6,11].

Although early identification and diagnosis may be the best way to positively influence outcomes in young children with PAE [12], PAE appears to be under-recognized by early childhood practitioners [11]. The early identification of infants/toddlers with PAE is complicated in several ways. Challenges by multiple systems of care (i.e., health care, child welfare, early intervention, and infant mental health) to initiate and conduct universal screenings or identification processes for prenatal exposures may be, in part, inhibiting earlier referrals for FASD diagnosis [13]. In addition, not all infants/toddlers with PAE present with easy-to-recognize symptoms, such as characteristic physical findings (e.g., growth problems, FAS facial features, structural brain abnormalities, etc.) or severe neurodevelopmental/behavioral delays [14]. Instead, many infants/toddlers may have more subtle developmental or behavioral indicators [15,16] that are not as easily recognized by early childhood practitioners, thus translating into a missed opportunity for early identification and intervention. 

Caregivers, on the other hand, are often the first to raise concerns about their child’s development or behavior and can, therefore, serve as a critical step in identifying early delays or problems that may arise from PAE. Directing attention to caregiver-reported assessments, which constitute a valuable component of early childhood assessment, is one way to learn about caregiver concerns. Standardized caregiver-report measures that are commonly used to assess infant sensory processing and behavior provide a way for parents or guardians to examine and report child behaviors. They permit caregivers to express their concerns through rating scales and responses to open-ended questions. Although clinicians tend to focus their attention on rating scale outcomes, using open-ended questions to ask about parent concerns can often lead to responses that are more spontaneous and personal [17]. It is through open-ended questions that new or under-recognized concerns related to PAE may be uncovered, as caregivers provide responses in their own words and are not constrained by predetermined responses [18].

By the nature of their role and relationship, primary caregivers have a unique vantage point that makes them acutely aware of the day-to-day challenges faced by their child. This increased awareness makes caregivers a vital resource for identifying children whose development and behavior do not appear typical. Previous research with caregivers of children with PAE or FASD demonstrates caregivers’ vigilance to variations in their child’s development and behavior. In a study of 1400 structured interviews of caregivers raising children impacted by PAE, the outcomes confirmed caregivers were highly preceptive in differentiating the cognitive and behavioral challenges of children across the continuum of FASD [19]. In a second study on the foster mothers of children (ages 2–16 years), a multitude of problems was reported, including concerns related to child cognition, behavior management, and coping with the daily realities of life [20]. Likewise, a third study describing the lived experiences of eight birth mothers of a child/ren with FASD (8–30 years) reported cognitive concerns (i.e., problems with attention, comprehension, and memory) and problem behaviors (i.e., excessive crying or no crying, hyperactivity, aggressiveness), in addition to health issues and delayed developmental milestones [21]. A fourth study emphasized the concerns faced by caregivers, including FASD-related stigma, family stress, and a lack of knowledge by professionals [22]. Finally, in a study by Pruner et al. [8], caregivers were asked to reflect on the challenges faced by their children with PAE during their first three years of life. Caregivers reported a diversity of concerns spanning across all domains of development and further reflected on how early interventions met (or did not meet) those needs. Collectively, these studies recognize caregivers’ valuable observation skills and insights into their children’s developmental needs. In accordance with this, the Academy of Pediatrics recommends that health professionals ask about and attend to caregiver concerns as a first step toward the developmental surveillance of infants and young children [23,24].

The recent literature has emphasized a need for a strengths-based approach to assessment and intervention regarding children with FASD [25,26]. Guiding caregivers to identify child strengths during a clinical encounter can extend benefits to both caregivers and practitioners [27]. A strengths-based approach can offer parents a sense of hope, alleviate child-related stress, and strengthen parenting capacity [28,29,30]. When early childhood practitioners appreciate the variety of child strengths identified by caregivers in this population, it may enable them to recognize and celebrate these assets more easily and in partnership with caregivers. In a parallel process, a strengths-based approach can enhance the bond between the practitioner, the caregiver, and the child, thus building effective working relationships and perhaps reducing FASD-related stigma [28,31,32]. To this end, 20 years of caregiver surveys of patients diagnosed with FASD at the University of Washington Fetal Alcohol Syndrome Diagnostic and Prevention Network (FASDPN) clinic confirmed that caregivers were highly satisfied with the strength-based approach to assessment and intervention [33].

Understanding the types of strengths and positive attributes of infants/toddlers with PAE from the perspective of their caregivers can inform strengths-based interventions. While an extensive amount of research has documented the challenges and impairments experienced by individuals with FASD across their lifespan, less research has focused on identifying strengths at any age [26,34]. Four studies were identified that described caregiver perceptions of child strengths. One study found that the caregivers of children aged 5–21 years recognized many positive traits (i.e., friendliness, hard-working, compassion, etc.) and abilities (e.g., artistic, athletic, etc.) in their child [35]. A second study identified the relative strengths in personal selfcare and household chore activities compared with other adaptive skills for children ages 5–8 years [36]. Third, the caregivers of school-age children reported a range of personal strengths in the context of students’ educational experiences, describing their child as being artistic, having strong verbal skills. or having good work habits [37]. In a fourth study, all caregivers were eager to share what they enjoyed most about their child during the early intervention period, including moments of affection, love, and laughter [8]. Notably, Olson and Montague [38] reported the strengths of young children with FASD based on informal reports, which “are filled with descriptions of how engaging, innocent, straightforward, amusing, curious and social young children with an FASD can be”. When taken together, these studies and informal reports highlight caregivers’ awareness of child strengths and their willingness to communicate these strengths to others. 

The present study was designed to address the following questions: (1) What are the concerns and strengths reported by caregivers regarding their infant/toddler (ages 7–42 months) with PAE? (2) From a descriptive perspective, do there appear to be patterns between caregiver types (birth parent, foster/adoptive parent, or other biological relatives) or child age (less than 2 years or 2 years and older) and the type or frequency of reported concerns and strengths? Examining caregiver-reported concerns may yield useful information regarding delays in child development or problem behaviors that warrant the attention of practitioners, signal the need for diagnostic referral, and/or lend important insight into the impact of these concerns on families. Understanding how child age or caregiver status can influence the reporting of concerns may facilitate a more targeted approach for practitioners when inquiring about caregiver concerns and for knowing what kinds of child development information and education certain families might need. In addition, the identification of child strengths and positive characteristics can provide opportunities to enhance parent–child interactions, incorporate these strengths into interventions, and help build caregiver-practitioner partnerships. 

## 2. Materials and Methods

The data for the current study were collected as part of a larger retrospective chart review of diagnostic assessment data from 125 infants/toddlers seen at the University of Washington FASDPN clinic between 2009 and 2019. This clinic does not require patients to present with a concern or delay, only a confirmed PAE at any level. Two linked studies were generated from this chart review, including (1) a descriptive study that examined the developmental, sensory processing, and behavioral outcomes of infants/toddlers with PAE [5]; and (2) the current study, which described caregivers’ early concerns and perceptions of their infant/toddlers’ strengths, based on data from two standardized caregiver questionnaires, the Infant/Toddler Sensory Profile (ITSP), and the Child Behavior Checklist/1½-5 (CBCL). All study activities were conducted with the University of Washington Human Subjects approval and caregiver consent at the time of their child’s FASD diagnostic evaluation.

### 2.1. Participants

Caregivers were included in this study if their infant/toddler met the inclusion criteria for the prior study [5] and they completed the ITSP questionnaire and/or the CBCL as part of their child’s FASD diagnostic evaluation. Child inclusion criteria for the first study were as follows: (1) age 1 month to 3.5 years—at the time of their FASD diagnostic evaluation; (2) received one of the following FASD 4-Digit Code diagnostic classifications (diagnostic categories A–C, D–J) reflecting the full continuum of outcomes observed among individuals with prenatal alcohol exposure (a) fetal alcohol syndrome (FAS; A,B) or partial fetal alcohol syndrome (PFAS; C), (b) static encephalopathy/alcohol exposed (SE/AE; E,F), (c) neurobehavioral disorder/alcohol exposed (ND/AE; G,H), (d) sentinel physical findings/alcohol exposed (I), or (e) no physical findings or central nervous system (CNS) abnormalities detected/alcohol exposed (Normal CNS/AE; (J) (see [39] for details about the FASD 4-Digit Diagnostic Code and [5] for detailed demographics of the prior study sample); and (3) had complete data on at least two domains of the Bayley Scales of Infant and Toddler Development (Bayley-III, [40]). Standardized parent questionnaires were completed by the primary caregiver prior to the scheduled diagnostic clinic date. Time, effort, or other demands placed on a caregiver may have resulted in some caregiver-report measures (i.e., Bayley-III Social-Emotional and Adaptive Behavior domains, ITSP, and CBCL) not being fully completed.

### 2.2. Measures

The data for this study were collected as part of a standard intake and diagnostic process for the FASDPN diagnostic clinic visit. The measures used for this study are described below.

*Infant Toddler Sensory Profile (ITSP [41]).* The ITSP is a 48-item caregiver questionnaire that measures sensory modulation abilities in daily life for infants/toddlers (7–36 months). Caregivers rate the frequency of infant/toddler sensory behaviors on a 5-point Likert scale. Caregivers also have the opportunity to respond to two open-ended questions: “What do you see as your child’s strengths?” and “What are your concerns?”. It is relevant to note that infants/toddlers older than 36 months were administered the Short Sensory Profile (SSP; [42]), which does not have open-ended questions as part of the questionnaire. Therefore, caregivers of infants/toddlers older than 36 months were included in this study if they completed the Child Behavior Checklist 1½–5 years only.

*Child Behavior Checklist 1½–5 years (CBCL [43]).* The CBCL is a 100-item caregiver questionnaire used to identify a range of emotional and behavioral problems in young children ages 1.5–5 years. Caregivers use a rating scale to determine the presence or absence of emotional and behavioral problems based on the preceding 2 months. Caregivers also have the opportunity to respond to two open-ended questions: “What concerns you most about your child?” and “Please describe the best things about your child”.

### 2.3. Descriptive Information for Participants

Three salient features from the FASD 4-Digit Diagnostic Code used in this study are briefly described below. See [39] for a more comprehensive description of these diagnostic features.


*The FASD 4-Digit Diagnostic Code generates the following clinical diagnoses. FAS, PFAS, SE/AE, ND/AE, Sentinel Physical Findings/AE; No Physical or CNS Abnormalities/AE.*


*Central Nervous System (CNS) Functional Rank.* Rank 1 = no dysfunction; Rank 2 = mild-to-moderate dysfunction; Rank 3 = severe dysfunction [39]. CNS functional ranks 1–3 document the severity of CNS dysfunction and are based on brain function (executive function, memory, cognition, social/adaptive skills, academic achievement, language, motor, attention, and activity level) assessed by an interdisciplinary team using standardized psychometric tools.

*Postnatal Risks.* Rank: 1 = no risk; 2 = unknown risk; 3 = some risk; and 4 = high risk) [39]. Individuals with PAE often present with a multitude of postnatal risks that could also be adversely impacting their development. Postnatal risk factors documented in the FASDPN database include perinatal complications, number of home placements, physical and/or sexual abuse, neglect, and trauma. The ranking is determined by clinical judgment at the time of the FASD evaluation and is based on available records and caregiver or other reports on intake forms and/or clinical interviews.

### 2.4. Data Analysis Plan

This study used a directed content analysis approach [44] to identify, categorize, and describe all instances of concerns and strengths reported by caregivers at the time of their child’s diagnostic evaluation. The directed approach begins with a framework for collecting and analyzing the data but allows for new insights to emerge through a process of inductive category development [43]. In this study, researchers aimed to validate an existing framework (domains commonly assessed in early childhood) in a new context (describing caregivers’ concerns and strengths). When the data did not fit into the existing framework, new categories were added to capture all possible instances of caregiver concerns and strengths [44,45].

The written responses to the two questions from the ITSP and two questions from the CBCL were extracted verbatim, excluding any identifying information. A coding system was developed that had multiple levels. First, responses were separated into two groups based on question type—concerns versus strengths/best things. Next, the researchers read all the caregiver comments related to concerns and organized the responses into broad categories. These broad categories arose from the data to reflect general areas of function or development (i.e., Development, Behavior, General/Medical, and Caregiving). After the data related to concerns were sorted, the data within each broad category were analyzed further to create subcategories reflecting the different examples of concerns within each broad category. Some of these subcategories were based on domains commonly assessed in early childhood or contained within the ITSP or CBCL measures, while others arose from the data (these are identified in Appendix A). A similar analysis was conducted with the strengths/best things data, with the broad categories being Development, Personality Traits, Interests, and Caregiving, and the subcategories within each of these are reported in Appendix B.

Two researchers (MP, JM) separately coded 50% of caregiver responses using the initial coding systems. They compared their results, and any discrepancies with how well the categories fit the data were discussed, and adjustments to the category structure were made. This process was repeated until consensus was reached, and all of the data total responses were coded. Once the coding systems were finalized, the first author coded the remaining responses.

Frequency counts for each coded category were calculated. Responses that were left blank or completed with statements such as “no concerns at this time” or “none” were also tracked. When a response contained multiple words or phrases that were suggestive of a concern/strength, it was only coded once. For example, the description “my child is extremely social, charming and loves other kids”, was coded under Sociability one time. On the few (*n* = 10) occasions that a response fell under two categories, the response was coded twice. For example, the description “doesn’t seem to understand” was coded under the two categories of Cognitive and Language concerns because the reason for the comprehension problems was not specified (i.e., whether it was a cognitive or language problem). Another example, “my child is easily over-stimulated”, was coded under the two categories of Regulatory and Sensory Processing because of the overlapping nature of this concern. When a caregiver completed both measures and shared similar concerns or strengths on both questionnaires, the responses were coded only one time. As a last step, the quantitative data were descriptively compared across age groups (i.e., <2 years and ≥2 years of age) and caregiver status (i.e., biological parent, foster/adoptive parent, other biological family).

## 3. Results

Records from 117 caregivers of infants/toddlers with PAE (ages 7–42 months) met the inclusion criteria for this study. Of these caregivers, 32% were birth parents, 44% were foster/adoptive parents, and 25% were extended relatives of the child (e.g., grandparent, aunt). An overwhelming majority of the sample (91%) presented with at least some level of postnatal risk in addition to their PAE. A total of 80% percent of caregivers in the sample completed the ITSP (7–36 months), while 57% completed the CBCL (1.5–3.5 years). See Table 1 for participant characteristics.

### 3.1. Concerns Identified

The coding process used for this study generated a list of 19 unique concerns expressed by caregivers (Appendix A). Caregivers reported an average of 2.5 concerns per child, ranging from 0–7 concerns per child. A total of 293 concerns were reported across the study sample. The five most frequently reported concerns were related to developmental and behavioral challenges and included aggressive behavior (27%), language/communication (22%), sensory processing (21%), internalizing problems (19%), and regulation (18%). Twenty-four (24%) caregivers did not report a concern for their child for either measure. The proportion of caregivers who did not report a concern was comparable across all three caregiver groups. The frequency of reported concerns across categories is presented in Table 2.

Concerns were explored across age groups. For younger infants/toddlers (<2 years), the caregiver concerns expressed most often were aggressive behavior (25%; *screams at high pitches; extreme temper*), sensory processing behaviors (24%; *sensitivity to sounds, lights, and clothes*), and motor skills (20%; *isn’t sitting up on his own*). For older infants/toddlers (≥2 years), caregivers reported the most concerns with language/communication skills (29%; *slow speech; doesn’t talk in more than three-word sentences*), aggressive behavior (29%; *can throw a fit that lasts for some time*) and internalizing problems (19%; *whiny, fussy, and sudden mood changes*). Notably, aggressive behaviors were reported with the most frequency across both age groups.

Concerns across caregiver status were also explored. Birth parents had the most concerns for PAE and other drug exposures (23%; *he was born addicted*), aggressive behavior (20%; *abusive*), and language/communication (18%; *worried about speech development*). The most common concerns reported by foster/adoptive parents were aggressive behavior (34%; *head butts, pulls out gobs of her own hair and even tries to pull mine out*), language /communication (25%; excessively repeats herself, doesn’t talk in more than three-word sentences), and sensory processing behaviors (26%; *has extremes in responses to stimuli*). The top concerns noted among other biological family members were language /communication (27%; *speech; constant chatter*), aggressive behavior (27%; *tantrums that are hard to calm down from*), and sensory processing (22%; *becomes inconsolable as soon as caregiver…introduces new sensation*). Aggressive behaviors were a top concern that was common across all three caregiver types.

### 3.2. Strengths Identified

The coding process generated a list of 20 unique strengths (Appendix B). Caregivers identified an average of 3.0 perceived strengths per child, ranging from 0–7 coded strengths each. A total of 352 strengths were coded across the study sample. The most frequent strengths or best things reported were reflective of personality traits: happiness (33%), sociability (30%), love/loving (28%), and curiosity (26%), and for developmental competencies, cognitive ability (22%). In contrast, strengths related to adaptive behavior (1%), eating/feeding (3%), and regulation (3%) were rarely reported. Strengths in the child’s interests (14%) and caregiver experience (3–5%) categories were also endorsed less frequently. A total of 25% of caregivers did not report a strength for either measure. The proportion of caregivers who did not report a strength was comparable across all three caregiver groups. Table 3 shows the frequency of reported strengths across categories.

Perceived strengths were explored across two age bands. For infants/toddlers (<2 years), caregivers frequently reported the following personality traits: curiosity (37%; *observant; curious*), happiness (31%; *very happy; happy most of the time*), and love/loving (29%; *child is lovable*). For older infants/toddlers (2–3.5 years), many caregivers described their child as happy (35%; *very happy girl*), social (35%; *loves to interact with me and other children*) and love/loving (30%; *very loving*).

Perceived strengths were also explored across caregiver status. Birth parents reported the most child strengths in the categories of happiness (27%; *happy baby*), love/loving (24%; *she is so loving*), and sociability (21%; *he’s a charmer*). Likewise, the most common strengths reported by foster/adoptive parents were happiness (43%; *she brings a lot of happiness to our lives*), sociability (32%; *friendly and outgoing*), and love/loving (27%; *loves her siblings*). The strengths expressed most often by other biological family members were love/loving (37%; *loving boy*), sociability (33%; *her smile and ability to get along with others*), and curiosity (33%; *tries everything, observant*). Sociability and love/loving were perceived strengths common across all three caregiver types.

## 4. Discussion

In this retrospective study of clinical data, the caregivers of infants/toddlers with PAE described a diversity of concerns and strengths in the context of two developmental questionnaires administered as part of their child’s FASD diagnostic evaluation. Our primary findings were that (a) caregivers’ predominate concerns fell into the categories of aggressive behavior and language/communication, while sensory processing and internalizing behaviors were also commonly reported; (b) the caregiver-perceived strengths spanned across numerous categories, with positive personality traits related to happiness, sociability, and love expressed most often, and (c) the type and frequency of reported concerns and strengths were relatively consistent across age and caregiver status. The findings from this study recognize the value of caregivers’ perspectives and offer an important reminder to practitioners that infants/toddlers with PAE and their caregivers have strengths that can be harnessed, in addition to a range of challenges that must be addressed.

Caregivers reported a broad array of concerns, reflecting the diversity of neurodevelopmental and behavioral outcomes known to be associated with PAE during early childhood [46,47]. Aggressive behaviors such as kicking and screaming, head banging, and prolonged temper tantrums raised the most concerns across both age groups and were relatively consistent across caregiver groups. Previous studies that examined behavior functioning found greater negative effects among infants [1] and preschool-age children with PAE [48,49], as well as a difficult temperament [50] and conduct-based problems [51] among preschoolers. Caregivers’ frequent concerns about language/communication are substantiated by studies that described delayed language abilities in infants with PAE [52,53,54]. Atypical sensory processing behaviors and internalizing problems were commonly described by caregivers in this study and, likewise, have been reported in the literature among infants/toddlers with PAE [15,55,56,57]. Concerns related to selfregulation, such as difficulty soothing or sleep complaints, were also consistent with selfregulatory difficulties seen in this population [58].

Overall, it appears that caregivers raised concerns that correspond closely to the outcomes from the standardized measures of development, sensory processing, and behavior in the existing literature. The findings demonstrate that caregivers can be an important source of information regarding their children. Explicitly asking caregivers about their concerns may aid in the earlier identification of delays or problems that may arise from PAE, especially when incorporated into an early routine screening or comprehensive clinical assessment [23].

Although most caregiver-reported concerns were coded using the categories from the core developmental domains, ITSP, and CBCL scales, there were a few exceptions. For example, concerns related to parenting were identified by 15% of caregivers, yet the ITSP or CBCL did not explicitly prompt caregivers to consider these concerns. Along the same lines, other concerns (e.g., PAE and other drug exposures and physical and health problems) were reported that were not captured on the questionnaires caregivers had previously completed. While many caregivers’ developmental, behavioral, and clinical concerns would have been assessed and/or detected at the time of their child’s FASD diagnostic evaluation, it is possible that the concerns related to parenting may have been missed if it were not for the open-ended questions. Caregivers who are struggling with the day-to-day stressors of raising an infant/toddler with PAE often require additional support to engage in sensitive and responsive parenting. With this goal in mind, these findings offer a reminder to practitioners that a combination of assessment approaches is needed to ensure that caregivers have more than one avenue to share concerns. Practitioners need to understand the complex problems facing caregivers, as well as their priorities and desired outcomes, to design treatment plans that are congruent with caregiver goals.

An overwhelming majority of caregivers (88%) shared one or more strengths/best things about their infant/toddler. Happiness, sociability, and love were the strengths reported most often, suggesting many caregivers believed these personality traits were worth knowing about and communicating with others. The finding that caregivers endorsed happiness and sociability among their top two strengths is consistent with a large meta-analysis examining the benefits of frequent positive affect in individuals across multiple life domains, which found happiness to be positively correlated with sociability [59]. Based on previous research on parents of neurotypical children (ages 3–9), love was also a frequently endorsed character trait [60]. For early childhood practitioners, this is useful information for initiating a working relationship with all types of caregivers, regardless of the child’s age. Recognized strengths, such as happiness, sociability, and love, may be perceived as a healthy indicator of parent–child connectedness and a starting point for noticing and exploring these perceptions, which is especially important for family-centered care and relationship-focused approaches [61,62]. Alternatively, when caregivers struggle to identify strengths in their child, this may signal to practitioners a need to promote attuned and positive exchanges between caregiver and child. A strengths-based approach is particularly important given the stigma associated with FASD. Both the biological, as well as nonbiological parents of children with PAE experience stigmatization when they are perceived as responsible for their child’s negative behavior or delayed development [32]. Cultivating child and caregiver strengths is in alignment with guiding principles of early intervention practice [9], as well as practice guidelines specific to families impacted by substance use [63]. Elevating strengths-based approaches for young children with known or suspected PAE and focusing on child strengths may work towards reducing stigma among caregivers and foster earlier identification and intervention as critical protective factors as early in life as possible [64].

Caregivers identified a few characteristics in both a positive and negative light. For example, the personality trait “zest” was reported by 16% of caregivers when they used phrases such as “my child’s personality is larger than life” or “he’s a firecracker”. In contrast, it appeared some caregivers perceived their child’s high energy and excitement as a problem related to attention or hyperactivity (i.e., she is very driven…very hyperactive, causing her to fall or run into things often). Furthermore, caregivers reported sociability as frequent strength, yet they also identified language/communication problems as a frequent concern. While concerns related to the use of language/communication are fundamentally different than the personality trait of sociability, practitioners can play a role in leveraging a child’s strong social skills toward the goal of developing language and communication skills. Indeed, building on strengths to compensate for child difficulties is a central intervention principle used with families raising children with FASD [65].

The following are study limitations. Since this was a retrospective chart review, the data were limited to the written responses reported on the assessment forms. As such, we were not able to probe for further detail or ask for clarification about any of the caregiver responses or follow up with those who did not respond. Responses to these questions were optional, and thus there may be bias or differences among caregivers who responded to the open-ended questions compared with those who did not. Caregivers of all types reported strengths and challenges on both measures; however, we cannot generalize results to all young children with PAE and caregivers due to the inherent limitation of a clinical sample. In order to gain a more thorough understanding of caregivers’ concerns and child strengths, future research should use a more systematic approach, such as guided interviews or focus groups that pose similar open-ended questions, with the opportunity to ask directed questions about strengths and challenges so that all participants have an equal ability to respond.

## 5. Conclusions

Caregivers identified concerns that warrant the attention and action of early childhood providers, demonstrating their attunement to the early challenges faced by their children. Caregivers also perceived their children to have many strengths across multiple areas. These findings suggest the importance of understanding the range of concerns and strengths that caregivers perceive in their day-to-day interactions with their children, which can enhance the development of family-centered interventions, strengthen parent–child connectedness, and build effective working relationships between early childhood practitioners and families impacted by PAE.

## Figures and Tables

**Table 1 children-10-00544-t001:** Demographic and clinical characteristics of the 117 participants.

Caregiver and Child Characteristic	N	(Valid %)
Respondent		
Biological mother	34	(29.1)
Biological father	3	(2.6)
Other biological family member	29	(24.7)
Foster parent	44	(37.6)
Adoptive parent	7	(6.0)
**Total caregiver sample size**	**117**	
Completed ITSP	94	(88.7)
Eligible to complete ITSP	106	
Completed CBCL	67	(82.7)
Eligible to complete CBCL	81	
Completed both ITSP & CBCL	54	(46.2)
Age of child described (years)		
0.5 to 0.99	14	(12.0)
1–1.99	45	(38.4)
2–2.99	44	(37.6)
3–3.5	14	(12.0)
**Mean (SD)**	1.99	(0.78)
Sex of child at birth		
Female	60	(51.3)
Male	57	(48.7)
FASD Diagnosis (Diagnostic category)		
FAS	3	(2.6)
PFAS	4	(3.4)
SE/AE	13	(11.1)
ND/AE	72	(61.5)
Sentinel physical findings/AE	5	(4.3)
No sentinel physical findings or CNS abnormalities detected/AE	20	(17.1)
CNS Functional Rank		
Rank 1, no dysfunction	27	(23.1)
Rank 2, moderate dysfunction	86	(73.5)
Rank 3, severe dysfunction	4	(3.4)
Postnatal Risk: Rank		
1. No risk	10	(8.5)
2. Unknown risk	1	(0.9)
3. Some risk	69	(59.0)
4. High risk	37	(31.6)

*Notes:* Infant toddler sensory profile 7–36 months (ITSP); child behavior checklist 1.5–5 years (CBCL); fetal alcohol spectrum disorder (FASD); fetal alcohol syndrome (FAS); partial FAS (PFAS); static encephalopathy/alcohol-exposed (SE/AE); neurobehavioral disorder/alcohol-exposed (ND/AE).

**Table 2 children-10-00544-t002:** Prevalence of reported concerns.

		Age Bands		Caregiver Type		
	Total Sample (*n* = 117)	<2 Years (*n* = 59)	≥2 Years (*n* = 59)	Birth Parent (*n* = 33)	Foster/Adoptive Parent (*n* = 56)	Other Biological Family (*n* = 27)
Category	*n* (Valid %)	*n* (Valid %)		*n* (Valid %)		
**Developmental Concerns**						
Overall development	12 (10.3)	10 (16.9)	2 (3.4)	2 (5.7)	7 (12.5)	3 (11.5)
Cognitive	14 (12.0)	5 (8.5)	9 (15.5)	3 (8.5)	7 (12.5)	4 (15.4)
Language/Communication	**26 (22.2)**	9 (15.3)	**17 (29.3)**	**6 (18.2)**	**14 (25.0)**	**7 (26.9)**
Motor	13 (11.1)	**12 (20.3)**	1 (1.7)	5 (14.3)	4 (7.1)	3 (11.5)
Social-emotional						
Regulation	**21 (17.9)**	11 (18.6)	10 (17.2)	1 (2.9)	13 (23.2)	1 (3.8)
Attachment	7 (6.0)	3 (5.1)	4 (6.9)	1 (2.9)	2 (3.8)	1 (3.8)
Adaptive Behavior	9 (7.7)	4 (6.8)	5 (8.6)	1 (2.9)	4 (7.1)	3 (11.5)
Sleep	9 (7.8)	7 (11.9)	2 (3.4)	1 (2.9)	4 (7.1)	1 (3.8)
Eating/feeding	15 (12.8)	9 (15.3)	6 (10.3)	2 (5.7)	8 (14.3)	2 (7.7)
**Behavior Concerns**						
Internalizing problems	**22 (18.8)**	11 (18.6)	**11 (19.0)**	2 (5.7)	13 (23.2)	5 (19.2)
Externalizing problems						
Attention problems	12 (10.3)	7 (11.9)	5 (8.6)	5 (14.3)	4 (7.1)	5 (19.2)
Aggressive behavior	**32 (27.4)**	**15 (25.4)**	**17 (28.8)**	**7 (20.0)**	**19 (33.9)**	**7 (26.9)**
Sensory Processing	**24 (20.5)**	**14 (23.7)**	10 (17.2)	4 (11.4)	**14 (25.0)**	**6 (22.2)**
Behavioral inflexibility	11 (9.4)	4 (6.8)	7 (12.1)	1 (2.9)	8 (14.3)	2 (7.7)
Safety awareness	11 (9.4)	4 (6.8)	7 (12.1)	1 (2.9)	5 (9.4)	4 (15.4)
**Child Concerns (in general)**						
PAE & Other Drug Exposures	16 (13.7)	9 (15.3)	7 (12.1)	**8 (22.9)**	4 (7.1)	3 (11.5)
FAS Physical Findings	10 (8.5)	6 (10.2)	4 (6.9)	1 (2.9)	4 (7.1)	5 (19.2)
Physical or health problems	12 (10.2)	9 (15.3)	3 (5.2)	3 (8.5)	7 (12.5)	2 (7.7)
**Caregiving Experience**	17 (14.5)	6 (10.2)	7 (12.1)	5 (14.3)	8 (14.3)	3 (11.5)
**No concerns reported**	28 (23.9)	16 (27.1)	13 (22.4)	10 (30.3)	11 (31.4)	7 (26.9)

*Notes*. Bolded numbers indicate the top 5 (total sample) or the top 3 concerns (age and caregiver categories).

**Table 3 children-10-00544-t003:** Prevalence of reported strengths or best things.

		Age Bands		Caregiver Type		
	Total Sample (*n* = 117)	<2 Years (*n* = 59)	≥2 Years (*n* = 58)	Birth Parent (*n* = 33)	Foster/Adoptive Parent (*n* = 56)	Other Biological Family (*n* = 27)
Category	*n* (Valid %)	*n* (Valid %)		*n* (Valid %)		
**Developmental Competencies**						
Cognitive	**26 (22.2)**	12 (20.3)	13 (22.4)	5 (15.1)	14 (25.0)	6 (22.2)
Language/Communication	11 (9.4)	7 (11.9)	4 (6.9)	1 (3.0)	8 (14.3)	2 (7.4)
Motor/Movement	12 (10.3)	7 (11.9)	5 (8.6)	3 (9.1)	5 (8.9)	4 (14.8)
Social-emotional						
Regulation	3 (2.6)	3 (5.1)	0	1 (3.0)	2 (3.6)	0
Attachment	12 (10.3)	9 (15.3)	3 (5.2)	3 (9.1)	8 (14.3)	1 (3.7)
Adaptive Behavior	1 (0.9)	1 (1.7)	0	0	0	1 (3.7)
Eating/feeding	4 (3.4)	3 (5.1)	1 (1.7)	1 (3.0)	1 (1.8)	2 (7.4)
**Personality Traits**						
Happiness	**38 (32.5)**	**18 (30.5)**	**20 (34.5)**	**9 (27.3)**	**24 (42.9)**	5 (18.5)
Love/loving	**33 (28.2)**	**17 (28.8)**	**16 (29.6)**	**8 (24.2)**	**15 (26.8)**	**10 (37.0)**
Kindness	25 (21.4)	10 (17.0)	15 (25.9)	6 (18.2)	14 (25.0)	6 (22.2)
Affectionate	10 (8.5)	7 (11.9)	3 (5.2)	3 (9.1)	7 (12.5)	0
Humor	23 (19.7)	12 (20.3)	11 (19.0)	7 (21.2)	11 (19.6)	5 (18.5)
Sociability	**35 (29.9)**	15 (25.4)	**20 (34.5)**	**7 (21.2)**	**18 (32.1)**	**9 (33.3)**
Curiosity	**30 (25.6)**	**22 (37.2)**	8 (13.8)	7 (21.2)	13 (23.2)	**9 (33.3)**
Courage	15 (12.8)	8 (13.6)	7 (12.1)	4 (12.1)	5 (8.9)	6 (22.2)
Autonomy	12 (10.3)	9 (15.3)	3 (5.2)	3 (9.1)	4 (7.1)	6 (22.2)
Zest	18 (15.4)	6 (10.1)	12 (20.7)	6 (18.2)	6 (10.7)	6 (22.2)
Adaptable	19 (16.2)	13 (22.0)	6 (10.3)	4 (12.1)	13 (23.2)	2 (7.4)
**Child Interests**	16 (13.7)	7 (11.9)	9 (15.5)	4 (12.1)	10 (17.9)	1 (3.7)
**Caregiver’s Experience**						
Confidence with parenting	3 (2.6)	2 (3.4)	1 (1.7)	1 (3.0)	2 (3.6)	0
Appreciation for positive change	6 (5.1)	2 (3.4)	4 (6.9)	0	5 (8.9)	1 (3.7)
**No strengths reported**	23 (19.7)	12 (20.3)	11 (19.0)	7 (21.2)	10 (17.9)	5 (18.5)

*Notes*. Bolded numbers in the total sample column indicate the top 5 concerns, and bolded numbers in age and caregiver columns indicate the top 3 concerns.

## Data Availability

Data sharing is not available due to issues of privacy.

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
