# Peer review of "Concerns and Strengths: Caregiver Perceptions of Their Infant/Toddler with Prenatal Alcohol Exposure†"

_children, 2023, doi:10.3390/children10030544_

Round 1

Reviewer 1 Report

The paper presented here addresses a very important issue often overlooked by professionals working with young children with FASD - it highlights not only the developmental difficulties observed from birth, but also the strengths in the infant's functioning. Identification of these areas can facilitate appropriate therapeutic interventions.

Although early identification of developmental difficulties offers the best chance to improve the development, a significant group of young children with PAE appear to be under-recognised by early childhood practitioners. This is because they are a heterogeneous group, with a wide range of delays and abnormalities in development, sensory processing and/or emotional and behavioural functioning.

The authors hypothesised that caregiver sensitivity to subtle indicators of a child's neurobehavioural abnormalities is a key element in the early diagnosis of developmental abnormalities in a group of children with PAE. They carefully reviewed previous studies in which parents, both foster and biological, reported specific neurobehavioural difficulties of older children and infants, confirming the very high alertness of parents to developmental abnormalities in the first three years of a child's life.

In line with recent achievement of professionals concerning the childhood mental health, the authors also hypothesised that there is a great need for a strengths-based approach to assessment and intervention with children with FASD, as guiding caregivers to identify children's strengths during clinical assessment can increase benefits for both caregivers and practitioners, can provide parents with a sense of hope, alleviate child-related stress and strengthen parenting capacity, and reduce the stigma associated with a child's diagnosis of FASD. In contrast, a review of the literature found only four studies that described perceptions of child strengths from a carer's perspective - both based on formal and informal tools.

The authors set out to design a broad study to answer the questions:

1) What are the concerns and strengths reported by carers about their infants (aged 7-42 months)

2) Is the type of care provided/type of caregiver (biological parent, adoptive parent, other) or the age of the child (under 2 years, 2 years or older) related to the frequency of reported concerns and strengths?

The results presented in the reviewed paper are part of a larger retrospective study in the University of Washington Fetal Alcohol Syndrome Diagnostic and Prevention Network (FASDPN) from 2009-2019, with confirmed PAE.  The authors conducted an extensive review of diagnostic data from the medical charts of 125 infants/children with PAE and did a tremendous amount of comparative and interpretive work yielding very interesting and important results.

Reviewer comments:

Study participants

Caregivers were included in the study if their infant/child met the inclusion criteria and they completed the ITSP and/or CBCL as part of the FASD diagnosis (FASD 4-Digit diagnosis) in one of the following categories:

(a) Fetal Alcohol Syndrome (FAS) or Partial Fetal Alcohol Syndrome (PFAS),

(b) Static Encephalopathy / Alcohol Exposed (SE/AE),

(c) Neurobehavioural Disorder / Alcohol Exposed (ND/AE),

(d) Sentinel Physical Findings/Alcohol Exposed, or

e) No Physical Findings or Central Nervous System (CNS) Abnormalities Detected / Alcohol Exposed (Normal CNS/AE).

Question: For what reason were children with following diagnoses not included in the study:

Sentinel Physical Finding(S) / Static Encephalopathy (Alcohol Exposed) (SPF/SE/AE)

Sentinel Physical Finding(S) / Neurobehavioural Disorder (Alcohol Exposed) (SPF/ND/AE).

Results

Results were obtained to answer research question 1) What are the concerns and strengths reported by carers about their infants (aged 7-42 months)

The results obtained are given in % ranks, however, a more detailed statistical analysis for the results found is lacking, as the raw % ranks add little to the understanding of the frequency of changes observed and the understanding of correlations.

The value of the significance level is not given and therefore it is unclear according to which criteria the cut-off level for the % ranks in each column was adopted.

The lack of statistical analysis makes it difficult to compare results between groups of carers and between age groups, and therefore the research question was not answered: 2) Is the type of care provided/type of caregiver (biological parent, foster/adoptive parent, other) or the age of the child (under 2 years, 2 years or older) related to the frequency of reported concerns and strengths?

Author Response

Question: For what reason were children with following diagnoses not included in the study:

Sentinel Physical Finding(S) / Static Encephalopathy (Alcohol Exposed) (SPF/SE/AE)

Sentinel Physical Finding(S) / Neurobehavioral Disorder (Alcohol Exposed) (SPF/ND/AE).

Thank you, I can see why you asked this question. We did, in fact, include SPF/SE/AE and SPF/ND/AE in our sample, but we combined them into one group and labelled them SE/AE and ND/AE. We combined the groups in this way because the study’s focus was on brain function and combining across Categories E,F and G,H helped to increase the sample size (statistical power) of the diagnostic study groups.

Here is the more detailed distribution of our study sample.

N = 125

We have made the following revision in our manuscript in lines 167-174:

2) received one of the following FASD 4-Digit Code diagnostic classifications (diagnostic categories A-C, D-J) reflecting the full continuum of outcomes observed among individuals with prenatal alcohol exposure a) Fetal Alcohol Syndrome (FAS; A,B) or Partial Fetal Alcohol Syndrome (PFAS; C), b) Static Encephalopathy / Alcohol Exposed (SE/AE; E,F), c) Neurobehavioral Disorder / Alcohol Exposed (ND/AE; G,H), d) Sentinel Physical Findings/Alcohol Exposed (I), or e) No Physical Findings or Central Nervous System (CNS) Abnormalities Detected / Alcohol Exposed (Normal CNS/AE; J)

The results obtained are given in % ranks, however, a more detailed statistical analysis for the results found is lacking, as the raw % ranks add little to the understanding of the frequency of changes observed and the understanding of correlations. 

The value of the significance level is not given and therefore it is unclear according to which criteria the cut-off level for the % ranks in each column was adopted.

The lack of statistical analysis makes it difficult to compare results between groups of carers and between age groups, and therefore the research question was not answered: 2) Is the type of care provided/type of caregiver (biological parent, foster/adoptive parent, other) or the age of the child (under 2 years, 2 years or older) related to the frequency of reported concerns and strengths?

This is a good point brought up by the reviewer. Our aim was descriptive. We provided percentages based on frequency counts but chose not to do any comparative statistical analysis due to the exploratory qualitative nature of the study. We agree the question leads the reader to expect a statistical comparison and thus we opted to clarify the question. The following change has been made (lines 137-140):

“From a descriptive perspective, do there appear to be patterns between caregiver types (birth parent, foster/adoptive parent, other biological relative) or child age (less than 2 years, 2 years and older) and type or frequency of reported concerns and strengths?”.

In addition, another primary reason we focus on the descriptive analysis is that the responses to the clinical questionnaires were optional. We noted this as a study limitation in the original submission. In this revision, we clarified this point in the discussion (lines 423-429) and the suggestion that a more systematic study is in important next step. Further, to do a statistical comparison, it would have been necessary to collapse some categories to reduce the number of statistical comparisons made. We wanted to highlight the range of strengths and challenges reported.

Reviewer 2 Report

This article clearly sets out a content analysis of caregivers responses when asked about the strengths and challenges of their child with PAE. The authors skillfully build up the rationale, supported by existing literature, for the use of caregivers' unique observational skills. They highlight that a parent-driven approach can be a valuable resource to help in the potential under-identification of challenges experienced by children impacted by PAE, which may not picked up by standard quantitative measurement tools. This work can contribute to advancing our knowledge of this area and inform associated intervention and/or practitioner approaches, furthermore provide external validation for existing psychometric frameworks. The unique finding about challenges to caregiving as an area of concern is noteworthy. However, the elements of the current study that focus on elucidating child strengths, via caregiver reports, represent the more important contribution to the literature. Importantly, how this may provide opportunities for improvement to parent-child interaction and practitioner care and support, not usually picked up by standard measurement tools.

The article is very clearly written with a good flow. The background and rationale is well laid out and the method and results have an easy to follow style. My suggested revisions/clarifications are minor and are detailed below:

·       -   I suggest listing the two age groups in the abstract

·      -  Explain how you managed the data when caregivers completed both the CBCL and the ITSP. Is there potential for double coding if a parent reported the same or similar concerns or strengths on both forms?

Author Response

I suggest listing the two age groups in the abstract.

This change has been made on line 18 in the abstract.

Explain how you managed the data when caregivers completed both the CBCL and the ITSP. Is there potential for double coding if a parent reported the same or similar concerns or strengths on both forms?

When extracting written data from questionnaires on strengths for example, I made one column for “best things” (CBCL) and one column for strengths (ITSP). For caregivers that had responses to both questionnaires, I included both responses and placed them in the appropriate column (best things or strengths). In some instances, parents used the exact same description (i.e., loving, outgoing, jokester) for both questionnaires. We only coded these responses one time. In most other instances, parents shared similar-enough ideas (i.e., strong motor skills versus good motor skills or resilient versus keeps trying) and again, these ideas were only coded one time. Finally, some parents shared different ideas across assessments (i.e., interested in how things work versus loves to dance) and these of course, were coded separately. A similar process was used to code challenges from the CBCL and challenges from the ITSP.

We added the following revision to clarify this point in lines 261-262:

When a caregiver completed both measures and shared similar concerns or strengths on both questionnaires, the responses were coded only one time.

Reviewer 3 Report

I found this paper and the respective study interesting.

My first comment relates to the lack of a limitations section - I suggest this is critically iimportant and relates to the general comments below:

- you note early in your paper that there are challenges with professionals not recognizing PAE

- you state that caregivers have a better and earlier awareness of the challenges and developmental deficits for example

- I note that the majority of the respondents are not biological parents who ony represent 31% of your participants

- further, all participants are already part of a clinic focussed on diagnosis and therefore PAE has alr6eady been identified.

Addressing the limitaitons should attend to this in that it is easier for those who are not biological parents (in particular mothers) to ask for guidance, support and diagnosis as the stignma related to mother's use of alcohol during pregnancy is lessened. Although I understand the appraoch and the group of participants that was accessed it is important to achknowledge that this is a unique group who are likely more open to responding to the questions asked.

Author Response

My first comment relates to the lack of a limitations section - I suggest this is critically important and relates to the general comments below:

  • You note early in your paper that there are challenges with professionals not recognizing PAE.
  • you state that caregivers have a better and earlier awareness of the challenges and developmental deficits, for example.
  • I note that the majority of the respondents are not biological parents who only represent 31% of your participants.
  • further, all participants are already part of a clinic focused on diagnosis and therefore PAE has already been identified.

Addressing the limitations should attend to this in that it is easier for those who are not biological parents (in particular mothers) to ask for guidance, support and diagnosis as the stigma related to mother's use of alcohol during pregnancy is lessened. Although I understand the approach and the group of participants that was accessed it is important to acknowledge that this is a unique group who are likely more open to responding to the questions asked.

We would like to clarify that study limitations were reported in the original submission as seen in lines 422 – 437.

We appreciate the reviewer’s perspective and sensitivity towards stigma and the recognition of the limits of a clinical sample. In regard to these comments, we have two points. This proportion of birth parents in this sample is very similar to the proportion of birth parents in a larger clinical sample (Astley, 2010). We also recognize results cannot be generalized to all caregivers of children with PAE and have added this to the limitations (lines 430 – 433).

See this reference for more details: Astley, S. J. (2010). Profile of the first 1,400 patients receiving diagnostic evaluations for fetal alcohol spectrum disorder at the Washington State Fetal Alcohol Syndrome Diagnostic & Prevention Network. Journal of Population Therapeutics and Clinical Pharmacology, 17(1).

We mention stigma around PAE and speak to this in relation to both biological and non-biological parents. We highlight strengths and strengths-based approaches in this paper for this very reason. We also added a line to further emphasize these important points. See lines 403-412:

A strengths-based approach is particularly important given the stigma associated with FASD. Both biological, as well as non-biological parents of children with PAE experience stigmatization when they are perceived as responsible for their child’s negative behavior or delayed development [34]. Cultivating child and caregiver strengths is in alignment with guiding principles of early intervention practice [9], as well as practice guidelines specific to families impacted by substance use [63].  Elevating strengths-based approaches for young children with known or suspected PAE and focusing on child strengths may work towards reducing stigma among caregivers and foster earlier identification and intervention as critical protective factors as early in life as possible.

We have adjusted the limitation section (lines 425-437) to address this point as seen below.

“The following are study limitations. Since this was a retrospective chart review, data were limited to the written responses reported on the assessment forms. As such, we were not able to probe for further detail or ask for clarification about any of the caregiver responses or follow up with those who did not respond. Responses to these questions were optional, and thus there may be bias or differences among caregivers who responded to the open-ended questions compared with those who did not. Caregivers of all types reported strengths and challenges on the respective assessments; however, we cannot generalize results to all young children with PAE and caregivers due to the inherent limitation of a clinical sample. To gain a more thorough understanding of caregivers’ concerns and child strengths and to ensure every participant had an equal opportunity to contribute, future research could take the form of guided interviews or focus groups posing similar open-ended questions.